# From Continuous-Time Chaotic Systems to Pseudo Random Number Generators: Analysis and Generalized Methodology

**DOI:** 10.3390/e23060671

**Published:** 2021-05-26

**Authors:** Luciana De Micco, Maximiliano Antonelli, Osvaldo Anibal Rosso

**Affiliations:** 1Facultad de Ingeniería, Universidad Nacional de Mar del Plata (UNMdP), Juan B. Justo 4302, Mar del Plata B7608FDQ, Argentina; maxanto@fi.mdp.edu.ar; 2Instituto de Investigaciones Científicas y Tecnológicas en Electrónica (ICyTE), Juan B. Justo 4302, Mar del Plata B7608FDQ, Argentina; 3Consejo Nacional de Investigaciones Científicas y Técnicas (CONICET), Rivadavia 1917, Buenos Aires C1033AAJ, Argentina; 4Instituto de Física, Universidade Federal de Alagoas (UFAL), Maceió 57072-900, Brazil; oarosso@gmail.com

**Keywords:** PRNG, statistical properties, NIST, diehard, chaos, permutation entropy, permutation complexity

## Abstract

The use of chaotic systems in electronics, such as Pseudo-Random Number Generators (PRNGs), is very appealing. Among them, continuous-time ones are used less because, in addition to having strong temporal correlations, they require further computations to obtain the discrete solutions. Here, the time step and discretization method selection are first studied by conducting a detailed analysis of their effect on the systems’ statistical and chaotic behavior. We employ an approach based on interpreting the time step as a parameter of the new “maps”. From our analysis, it follows that to use them as PRNGs, two actions should be achieved (i) to keep the chaotic oscillation and (ii) to destroy the inner and temporal correlations. We then propose a simple methodology to achieve chaos-based PRNGs with good statistical characteristics and high throughput, which can be applied to any continuous-time chaotic system. We analyze the generated sequences by means of quantifiers based on information theory (permutation entropy, permutation complexity, and causal entropy × complexity plane). We show that the proposed PRNG generates sequences that successfully pass Marsaglia Diehard and NIST (National Institute of Standards and Technology) tests. Finally, we show that its hardware implementation requires very few resources.

## 1. Introduction

Many engineering applications require the utilization of random numbers, such as in the area of communication, encryption, codification, and modulation [1,2,3].

The use of chaotic systems, such as Pseudo-Random Number Generators, has recently grown because of the multiple advantages they present over stochastic algorithms [4,5,6,7]. In [8], the authors propose a five-step encryption algorithm. One of these parts is a chaotic systems module, where the system chooses between different number generators. In the conclusions, the authors highlight the importance of its randomness and present digital degradation as a subject to study. It is well known that both chaotic maps and continuous-time chaotic systems have internal correlations (they can be easily seen in 3D plots of their outputs) that prevent them from being used as PRNGs. The picture is even worse in the case of continuous-time chaotic systems that, unlike chaotic maps, present strong temporal correlations. In the literature, there are studies that promise to generate good PRNGs from continuous-time chaotic systems that can be used in cryptography and secure communications. In general, they fail in three main aspects: speed, randomness, and generality. These qualities are essential for any PRNG. A slow PRNG is useless for almost all applications, for example, for real-time encryption. To ensure security, the sequences have to pass statistical tests, such as Diehard or NIST. Otherwise, there exist tools capable of detecting the inner correlations and thus capable of breaking the security. Finally, generality, because a good chaotic PRNG should not depend on a single chaotic system. It should work with other chaotic systems [9].

In [10,11], the authors employ a skipping technique to enhance the randomness of the chaotic outputs (called self-cascading in [11]). Instead of iterating with the original map *f*, it uses its d-times iterated one $f^{d}$ [12]. Actually, this technique hides the correlations rather than destroying them. The higher the iteration used, the less the structures can be seen; however, the sequences still keep their internal correlations (lacks in randomness).

If the iterations are alternated between different maps, the method is called switching; this is also proposed in [11] (called hybrid-cascading there), in [13,14]. The limitation is that the maps must have common convergence domains, or at least common areas, which are not easy to find (lacks in generality).

Some works [15,16] iterate chaotic systems using floating point architecture and complex integration algorithms (such as Runge Kutta) and apply some type of post-processing or coding to eliminate the internal structures. Floating point operations, as well as complex integration algorithms, require many calculations. This, added to the post-processing calculations, limits the output speed (lacks in speed) and also requires lots of resources. Another frequently used method is to introduce external disturbances to the system. In these cases, the randomness of the final system turns out to be that of the disturbing system; unlike what may be expected, the randomness is not added, which is the case of [13,14].

A successful technique for obtaining random outputs from continuous time chaotic systems is the discarding method ([17,18,19,20], called the deep-zoom method in the latter). It basically consists of dismissing the most significant bits of each output, and it exploits the fact that chaos analytically relies on the infinitesimal depth of precision digits used. However, to maintain chaotic oscillation, they are forced to use a high number of bits (even floating point arithmetic) and complex temporal discretization methods. Furthermore, due to the internal and temporal correlations of these systems, a low number of bits for the PRNG can be taken (lacks in speed) at each iteration.

There are works that propose to generate random sequences by applying fractional calculus to existing chaotic systems or even using new defined fractional chaotic systems [21,22]. The novelty is that continuous-time systems of less than three dimensions can exhibit chaotic behavior. However, the typical internal structures of chaotic systems remain in the sequences generated by these systems. So, they are in the same situation as mentioned before. What is more, the procedure of generalizing the integer derivative and integral orders to real and complex numbers requires more calculations, and it is not clear if having fewer dimensions is an advantage; for example, in our approach, we take advantage of the three dimensions as we extract bits from the three variables, so to increase throughput.

Traditionally, continuous-time chaotic systems have not been the preferred choice over chaotic maps mainly due to the strong time correlation and the extra computations they require to perform the time-discretization.

There exist an ample variety of numerical algorithms to solve ordinary differential equations. Which of them to choose will depend on the final objective. Here, rather than generating what some researchers call the “true solutions”, the interest is to obtain the most random output while keeping the chaotic behavior. Meanwhile, it is desired to employ the least amount of resources in terms of hardware, so the simplest method would be preferable. The drawback is that, in general, the simplest numerical methods produce the trajectories to converge or tend to cycles with short periods.

Here, we analyze numerical integration algorithms looking for the one that (i) maximizes the randomness degree, and (ii) requires the fewest resources regarding its hardware implementation.

The time correlation is related to the integration step, Δt. The lower Δt, the more time-correlated the output would be. However, a large Δt could result in the system losing its chaotic behavior. Therefore, choosing the appropriate step Δt is not a trivial task. We propose a point of view where the continuous system becomes a discrete map, and the time step used is seen as an extra parameter of this new map. This enables us to characterize the system in terms of Δt and use statistical quantifiers as well as nonlinear tools to describe the dynamics’ evolution of the maps.

Our goal is to propose an extremely simple modification applied to the digitalized continuous-time chaotic system that keeps it oscillating and, at the same time, breaks the internal structures and the time correlation of the outputs, which allows us to apply the discarding method, but discarding a minimum number of bits, so as not to lose speed. Using the standard Marsaglia’s Diehard and NIST tests, we show that the resulting map can generate high-quality random numbers to ensure security. Finally, our method is general as it can be applied to any continuous-time chaotic system. We also present the resources needed by a hardware implementation in an FPGA board of the proposed PRNG and compare them with the ones of the original map, showing that the circuit complexity remains almost the same.

The rest of the paper is organized as follows. Section 2 presents the ordinary differential equations that concern us, briefly describes the methods that we consider for the time discretization of those systems and presents the proposed modification over the Euler method. Section 3 gives a short review of the quantifiers employed to characterize the maps’ chaoticity and randomness. In Section 4, the obtained results when applying the proposed methodology to the Rössler system are presented. There, we develop new maps that emerge from applying the numerical methods and the proposed modification. Finally, in Section 5, we draw some concluding remarks.

## 2. Continuous-Time Chaotic Systems

The general form of a typical continuous-time chaotic system is as follows:(1)$$\dot{u}=f\left(t,u\right)$$
where f(t,u) are nonlinear functions of time and the states variables *u*. Given an initial value $u\left(t_{0}\right)=u_{0}$, these systems have a determined evolution. However, there is no general formula to solve this kind of equation; in fact, most of these first-order differential equations cannot be analytically solved [23]. That is why particular time-dependent solutions are most often sought with numerical means. Thus, multiple techniques that approximate the output of the system have emerged. Among the ample range of possibilities for making such a job, the choice of one of them depends on several factors. When an exact reproduction of the continuous system dynamics is required, powerful numerical methods that involve pre-iterations and variable time-steps are mandatory. However, when using these systems, such as PRNGs, the criteria change for choosing which method to use changes. It switches to the ones that allow the output to meet the required properties along with the strong restrictions regarding the hardware implementation, i.e., minimize the required resources and latency, maximize throughput and operation frequency. Considering the above, we focus on fixed integration step methods. In this context, we evaluate the following three well-known numerical methods for the time-digitization of continuous-time systems.

### 2.1. Fourth Order Runge–Kutta Method (RK4)

The main idea of this method is the precalculation of stages at various points using samples of *f* to obtain the next step [24].
(2)$$\begin{array}{ccc} u_{t+\Delta t} & = & u_{t}+\frac{\Delta t}{6}\left(k_{1}+2k_{2}+2k_{3}+k_{4}\right), \end{array}$$
$$\left\{\begin{array}{c} k_{1}=\Delta tf\left(t,u_{t}\right),\\ k_{2}=\Delta tf\left(t+\frac{\Delta t}{2},u_{t}+\frac{k_{1}}{2}\right),\\ k_{3}=\Delta tf\left(t+\frac{\Delta t}{2},u_{t}+\frac{k_{2}}{2}\right),\\ k_{4}=\Delta tf\left(t+\Delta t,u_{t}+k_{3}\right), \end{array}\right.$$
where $u_{t}$ is the discrete-time state variable and Δt is the time step size.

### 2.2. Heun’s Method (HUN)

Heun’s method considers the tangent lines to the solution curve at both ends of the interval. This method requires two stages of calculation as follows:(3)$$\begin{array}{ccc} u_{t+\Delta t} & = & u_{t}+\frac{\Delta t}{2}\left[f\left(t,u_{t}\right)+f\left(t+\Delta t,\tilde{u_{t}}\right)\right],\\ \tilde{u_{t}} & = & u_{t}+\Delta tf\left(t,u_{t}\right) \end{array}$$

### 2.3. Euler’s Method (EUR)

Among all numerical procedures for solving ordinary differential equations with a given initial value, the simplest one is Euler’s method in which differentials are approximated by a trapezoid with base Δt, as Equation (Equation 4) shows. Euler’s method is a one-step algorithm; that is, in order to calculate the variables at the time t+Δt, it is only necessary to know the values at the previous instant.
(4)$$\dot{u_{t}}\approx \frac{u_{t+\Delta t}-u_{t}}{\Delta t},$$
where:(5)$$u_{t+\Delta t}=u_{t}+\Delta tf\left(u_{t}\right)$$

### 2.4. Modified Euler Proposed Method (EUR_MOD)

Choosing a large Δt would be believed to help de-correlate the output of the system and thereby improve its randomness. However, the largest Δt before the system loses its chaotic behavior is not big enough to break its temporal structures. There are numerous proposals to increase the randomness of chaotic systems [25,26,27]. In some of them, post-processing the outputs is proposed; however, this idea adds hardware and increases latency. Other works propose to disturb the system with external noise, to switch between one or more chaotic maps [28]. Then, complexity is added to the resulting circuit, but the achieved randomness of the final system is just that of the disturbance. As said, the objective is to destroy the temporal correlations while keeping the chaotic oscillation of the system. Furthermore, it is desired to minimize the hardware resources and increase throughput and speed. Our idea is to combine the time-digitalization process with the randomization one. Thus, we have selected Euler’s method as is the simplest one and thereby will require the least amount of hardware to be implemented. Based on Equation (Equation 5), with the idea of breaking the temporal structure, we apply the following modification:(6)$$\left\{\begin{array}{c} x_{t+\Delta t}=x_{t}+\Delta tf_{x}\left(x_{t},y_{t},z_{t}\right)+p_{1}z_{t}{\left(-1\right)}^{x_{t}\;mod\;2}\;\\ y_{t+\Delta t}=y_{t}+\Delta tf_{y}\left(x_{t},y_{t},z_{t}\right)+p_{2}x_{t}{\left(-1\right)}^{y_{t}\;mod\;2}\;\\ z_{t+\Delta t}=z_{t}+\Delta tf_{z}\left(x_{t},y_{t},z_{t}\right)+p_{3}y_{t}{\left(-1\right)}^{z_{t}\;mod\;2}\; \end{array}\right.$$
where $u_{t}\;mod\;2$ returns the remainder of a division after $u_{t}$ is divided by 2. It returns 1 if $u_{t}$ is odd, or 0 if it is even. The parameters $p_{1}$, $p_{2}$ and $p_{3}\in \left[0,1\right]$, so for the particular case where $p_{1}=p_{2}=p_{3}=0$, the map converges to $ROS_{EUR}$ map.

The modification consists of incorporating one extra term into each function. This term is simply another state variable multiplied by 0.5. That term will be added or subtracted from the function depending on the parity of the current state variable.

## 3. Quantifiers

The time-digitization of continuous systems that turns them into maps generates changes in their dynamics. Chaoticity, stochasticity, and mixing properties change, so the following tools are used to analyze them.

### 3.1. Maximum Lyapunov Exponent

A chaotic orbit (chaotic attractor) is aperiodic, meaning that it never repeats exactly itself, and the oscillation persists for a time tending to infinity. The attractor’s movements exhibit sensitive dependence on the initial conditions. This means that two trajectories that start very close, quickly diverge; thus, they will have very different futures. The practical implication of this is that long-term prediction becomes impossible, as small uncertainties are rapidly amplified. The separation δ(t) between two trajectories of the same system that initially differ $\delta_{0}$ evolves exponentially in the way of Equation (Equation 7):(7)$$\left||\delta \left(t\right)|\right|∼\left|\right|\delta_{0}\left|\right|e^{\lambda t}$$

Therefore, neighboring trajectories separate exponentially fast. The number λ is called the Lyapunov exponent. When this exponent is positive, it is said that the system has a time horizon beyond which the prediction fails at tolerance *a*. Actually, λ depends on the trajectory that is being studied. Therefore, it must be averaged over many points of the same trajectory to estimate its true value. In addition, each system has as many Lyapunov exponents as dimensions. The largest of them, known as the maximum Lyapunov exponent (MLE), is of special significance since a positive value indicates the possible existence of chaos [29,30]. Nevertheless, this is a necessary but not sufficient condition of chaoticity since a divergent system can have positive MLE. Therefore, for a system to be chaotic, in addition to having some positive Lyapunov exponent, it must have a bounded non-divergent trajectory in the phase plane.

### 3.2. Bifurcation Diagram

A bifurcation diagram allows studying the changes in the qualitative or topological structure of the trajectories of a dynamical system. It shows the visited values of a system as a function of a certain parameter. It allows differentiating areas of the parameter in which the system behaves like fixed points, periodic orbits, or chaotic attractors [31]. We can say that bifurcation occurs in a dynamical system when a small smooth change of a parameter causes a sudden ‘qualitative’ or topological change in the dynamical system’s behavior.

### 3.3. Probability Density Function (PDF)

The randomness quantifiers used here are functional of the PDF *P* associated with the data sequence under analysis. The determination of a PDF can be done using several different methods [32], and their applicability depends on particular characteristics of the data, such as stationarity, time series length, parameter variation, and level of noise contamination. The PDF and the sample space are inextricably linked so it is a nontrivial problem to obtain the optimal PDF to extract the desired information. Here, we have employed the Bandt and Pompe approach, and this PDF is able to satisfactorily show the temporal correlations of higher orders [33,34]. The delay method has been used to extract time causal information from the sequences. A delayed reconstruction in *D* dimensions is formed by the vectors $x_{n}$ given as:(8)$$x_{n}=\left(x_{n-(D-1)v},x_{n-(D-2)v},\cdots ,x_{n-v},x_{n}\right)$$
The lag or delay time *v* is the time difference in the number of samples (or in time units τ=vΔt) between adjacent components of the delay vectors. A good estimate of the lag time is very difficult to obtain. If τ is small compared to the internal time scales of the system, successive elements of the delay vectors are strongly correlated, whereas if τ is very large, successive elements are already almost independent. Among the existing proposals, we have adopted the first zero of the autocorrelation function of the signal as the τ value [30]. This algorithm to extract the Bandt–Pompe PDF has been widely addressed and described by previous works [35].

#### 3.3.1. Normalized Shannon Entropy

The well-known normalized Shannon entropy denotes the amount of “disorder” a system presents. It has been shown to be able to successfully characterize determinism and stochasticity of generated sequences [32]. This information theory quantifier is a functional of the probability density function and is defined by the normalized Shannon expression (Equation (Equation 9)):(9)$$H\left[P\right]=-\frac{\sum_{i=1}^{N}p_{i}\ln\left(p_{i}\right)}{\ln(N)};$$
where *N* is the number of elements of the alphabet. We denote permutation entropy ($H_{BP}$) as the result of applying the normalized entropy to the PDF proposed by Band and Pompe, which quantifies the causality of the symbolic series discarding amplitude information.

#### 3.3.2. Statistical Complexity Measure

A statistical complexity measure, denoted by *C*, is a general indicator of structure or correlation. This measure vanishes in the extreme ordered and disordered limits (“boundary conditions”). During the last decade and a half, different measures of statistical complexity have been proposed [36]. Here, we have adopted the functional form introduced by López Ruiz et al. [37] with the modifications advanced by Lamberti et al. [38], given by Equation (Equation 10), [39].
(10)$$C\left[P\right]=Q_{j}\left[P,P_{e}\right]H\left[P\right],$$
where $P_{e}$ is the uniform distribution, and $Q_{j}$ is the so-called “disequilibrium”, defined in terms of the extensive Jensen–Shannon divergence, which in turn induces a squared metric, [39] (Equation (Equation 11)).
(11)$$Q_{j}\left[P,Pe\right]=Q_{0}\left\{S\left[\frac{\left(P,P_{e}\right)}{2}\right]-\frac{S[P]}{2}-\frac{S[P_{e}]}{2}\right\},$$
where $Q_{0}$ is the normalization constant, Equation (Equation 12), and is obtained when the system is deterministic; that is, only one component of P is equal to one, and the remaining components are equal to zero:(12)$$Q_{0}=-2{\left[\frac{\left(N+1\right)}{N}\ln\left(N+1\right)-\ln\left(2N\right)+\ln\left(N\right)\right]}^{-1}.$$

This quantifier detects internal structures from the symbol source when it is applied to the Bandt and Pompe PDF; thus, we denoted permutation complexity $C_{BP}$ as the resulting quantity.

The juxtaposition in a two-dimensional graph of the quantifiers $H_{BP}$ and $C_{BP}$ has demonstrated to be particularly efficient to reveal properties of the underlying processes from some measurable or observable quantity, called causal Eentropy × complexity plane [40]. High values of $C_{BP}$ correspond to time series with immersed structures, which occurs with chaotic series. On the other hand, the point $C_{BP}=0$ and $H_{BP}=1$ are that of a sequence with no internal correlations. There are many relevant applications of the $H_{BP}\times C_{BP}$ plane; for example, in [34], Rosso et al. use this plane to discriminate between stochastic and chaotic series, in [41], the authors employ it as a tool for distinguishing songs, and Zunino and Ribeiro utilize it to discriminate image textures [42], just to mention a few.

### 3.4. Statistical Randomness Tests

For a sequence to be suitable to be used as PRNG, it is necessary to successfully pass statistical tests. Here, we employed NIST Statistical Test Suite and Marsaglia Diehard tests.

#### 3.4.1. NIST Statistical Test Suite

The NIST SP 800-22 test suite [43] consists of 15 statistical randomness tests that are applied to binary data stream files. It requires the size of each sequence length to be of the order $10^{3}$ to $10^{7}$. For each test, it yields *p*-values, and it also checks the proportion of passing sequences and the uniform distribution of the *p*-values.

#### 3.4.2. Marsaglia Diehard Tests

The 15 statistical tests that make up the Diehard battery should be applied independently over files of several million 32-bit integers. Their output is a statistical *p*-value. To evidence randomness, each test output should be uniformly distributed between 0 and 1. The tests should be repeated multiple times with different integer sets to demonstrate the robustness of outcomes.

## 4. Results

To show the proposed method, the well-known Rössler system is used here. This continuous-time chaotic system is defined by the following set of coupled ordinary differential Equations [44]:(13)$$\left\{\begin{array}{c} \dot{x}=-y-z,\\ \dot{y}=x+ay,\\ \dot{z}=b+z\left(x-c\right). \end{array}\right.$$

Applying the digitalizing methods mentioned in Section 2, the following maps, which include Δt as a new parameter, are obtained:$ROS_{RK4}$ map, Rössler system digitalized by the 4th order Runge Kutta method.$ROS_{HUN}$ map, Rössler system digitalized by the Heun method.$ROS_{EUR}$ map, Rössler system digitalized by the Euler method.$ROS_{EUR\_MOD}$ map, Rössler system digitalized by our proposed Euler modified method.

We have employed parameters a=0.2, b=0.2, and c=5.7 that assure chaotic behavior of the continuos-time Rössler system. In the case of the $ROS_{EUR\_MOD}$ map, the parameters $p_{1}=p_{2}=p_{3}=0.4$ were used unless specified otherwise. Since our objective is to utilize the systems as PRNGs, based on subsequent experiments, we have followed three main steps:First, we analyzed the chaotic behavior when the systems are digitalized in time; focusing on the impact on the dynamic of each discretization method and its dependence on Δt (Section 4.1). Therefore, we calculate the MLE [29] and bifurcation diagrams of the emerged maps. Note that at this point, we do not consider amplitude discretization of the systems. Therefore, we employ a floating-point arithmetic (IEEE 754 double-precision standard) for the calculations.The second step deals with the amplitude digitization effect (Section 4.2). Then, we analyze the statistical properties, focusing on achieving the highest randomness.Finally, we present the hardware implementation of the obtained PRNG that is based on the proposed modification to the system digitalized in time by Euler’s method and iterated using signed fixed-point architecture. We also show the resources needed to implement it in an FPGA board (Section 4.3).

### 4.1. Time Digitization Analysis

In all cases, the quantifiers are averaged over 100 surrogates starting at different initial conditions, a transitory of $8\times 10^{6}$ is first deleted, and the maps are then iterated $10^{6}$ times. The minimum Δt is iterated $10^{6}$ times, and for higher Δt, the iterations are decreased so as to cover the same attractor window time. In order to understand the maps’ behaviors, Figure 1 shows the 3D phase space for some Δt values of the $ROS_{HUN}$ map. There, it can be seen how attractors change and evolve. It is clearly shown that even though the continuous-time system attractor is blurred by the increase of Δt, new attractors appear, and these attractors may be even more chaotic than those of the continuous-time systems (depicted by their MLE value). As expected, smaller values of Δt reproduce an attractor closer to that of the continuous-time system; however, in many cases, they converge to short cycles or fixed points, losing their chaotic behavior.

#### 4.1.1. Topological Analysis

We have used Sprott’s method to calculate the MLE of the maps using Δt as a parameter, [45]. Figure 2a shows how the MLE varies with Δt in the case of the time-digitalizing Rössler system using the three methods. Each resulting map presents a different behavior regarding the existence of chaoticity with Δt as a parameter. As it may be supposed, the $ROS_{RK4}$ map (red line) seems to preserve the chaotic behavior for larger values of Δt, while the $ROS_{EUR}$ and $ROS_{HUN}$ maps (yellow and blue lines, respectively) behave similarly. In the case of the $ROS_{HUN}$ map, it presents an isolated range of Δt between ∼0.17 and ∼0.19, where the system behaves chaotically, and it also presents isolated low values of MLE for some time steps indicating low or no chaoticity. The $ROS_{EUR}$ map is the first one that losses its chaotic behavior for higher time steps. It is interesting to note that even though all three maps are derived from the same system for small values of Δt, the MLE does not show similar values. The reason may be the accumulated arithmetic errors that prevent following the continuous-time attractors. In addition, it can be seen that there are some cases where larger time steps present higher values of MLE.

In Figure 2b, it can be seen that the proposed modification increases the chaotic behavior of the system. The $ROS_{EUR\_MOD}$ map presents higher values of MLE than the $ROS_{EUR}$ map. To show the generality of the proposed method, Figure 3 shows the 3D phase of Rössler and Lorenz systems digitalized by Euler’s method ($ROS_{EUR}$ and $LOR_{EUR}$) in black, and their modified maps ($ROS_{EUR\_MOD}$ and $LOR_{EUR\_MOD}$) in gray. It can be seen how the proposed modification breaks the inner structures and temporal correlations of the sequences and keeps the chaotic behavior. This enables the retainment of more bits for the PRNG output and, in this way, increases the throughput.

Regarding the bifurcation diagram, we have built the diagrams using Poincaré maps, which is the intersection with a certain surface. Then, the bifurcation diagrams show all the visited values by the systems [31]. Figure 4 shows the bifurcation diagram of the $ROS_{HUN}$ map superimposed with the MLE (red line). It can be seen how the MLE is able to effectively predict the chaoticity of the map. Within the chaotic region, some isolated gaps that correspond to low chaoticity can be seen. These gaps match with low values of the MLE. It can be seen that from Δt∼0.105, it completely loses its chaoticity and it stays on a periodic cycle until Δt∼0.167. The darker areas of the chaotic region imply that the system, while being in the state of chaos, spends more time there than in the lightly shaded regions. The most interesting places inside that region are the “white spaces”, which have an important role in the transition to chaos. The “white regions” and their boundaries also show the instability of the initial conditions, another important aspect of the chaos.

#### 4.1.2. Statistical Analysis

Up to this point, we have only analyzed the chaoticity of the maps; however, we do not have information about the randomness that their outputs present. The applications in which these maps are intended to be used require that their sequences, in addition to being chaotic, have no internal structures and all their possible outputs appear in a balanced way. To evaluate this, we calculate the randomness quantifiers described in Section 3. Each quantifier has been averaged over 100 files. Every surrogate file starts with a different initial condition, a transitory of $8\times 10^{6}$ iterations was first deleted, and the maps were then iterated $10^{6}$ times.

To extract causal informati on by calculating $H_{BP}$ from these observations, we employ here *x* state variable, v=1, and D=6.

Figure 5 shows the plane $H_{BP}\times C_{BP}$ for the Rössler system time-digitalized with the three mentioned methods using different values of Δt. The continuous curves correspond to the boundaries of values for the statistical complexity, as a function of the value of the normalized Shannon entropy [38]. It can be seen that in the cases where the system is unmodified ($ROS_{RK4}$, $ROS_{HUE}$, and $ROS_{EUR}$ maps), the quantifiers remain in the same area, that is, strong correlations and poor balance of values. When the proposed modification is applied, the quantifiers move towards the area of chaotic maps. The output of the system slightly improves the balance of its values and also increases its inner correlations.

### 4.2. Amplitude Digitization Analysis

We iterate the maps using signed fixed-point architecture for analyzing the effect of amplitude digitalization [46]. As said, our goal is to develop a hardware-implemented PRNG, which is why we have selected fixed-point architecture and Euler’s discretization method because of the simplicity they mean in terms of hardware design (Equations (Equation 14) and (Equation 15)).

$ROS_{EUR}$ map:
(14)$$\left\{\begin{array}{c} x_{t+\Delta t}=x_{t}+\Delta t\left(-y_{t}-z_{t}\right)\\ y_{t+\Delta t}=y_{t}+\Delta t\left(x_{t}+ay_{t}\right)\\ z_{t+\Delta t}=z_{t}+\Delta t\left[b+z_{t}\left(x_{t}-c\right)\right] \end{array}\right.$$$ROS_{EUR\_MOD}$ map:
(15)$$\left\{\begin{array}{c} x_{t+\Delta t}=x_{t}+\Delta t\left(-y_{t}-z_{t}\right)+\frac{z_{t}}{2}{\left(-1\right)}^{x_{t}\;mod\;2}\;\\ y_{t+\Delta t}=y_{t}+\Delta t\left(x_{t}+ay_{t}\right)+\frac{x_{t}}{2}{\left(-1\right)}^{y_{t}\;mod\;2}\;\\ z_{t+\Delta t}=z_{t}+\Delta t\left[b+z_{t}\left(x_{t}-c\right)\right]+\frac{y_{t}}{2}{\left(-1\right)}^{z_{t}\;mod\;2}\; \end{array}\right.$$

We have employed words of *wl* bits, with *fl* bits to represent the fractional part in two’s complement arithmetic; this architecture is represented by S(*wl*,*fl*). Equation (Equation 16) outlines the data format used for each state variable.
(16)$$u_{t}=\overset{signed\;word\;(wl\;bits)}{\overset{︷}{\begin{array}{ccc} \underset{integer\;part}{\underset{︸}{b_{wl-1}b_{wl-2}\cdots b_{wl-(wl-fl)}}} & \underset{↑}{\cdot } & \underset{fractional\;part\;(fl\;bits)}{\underset{︸}{b_{fl-1}b_{fl-2}\cdots b_{0}}}\\ & fractional\;point \end{array}}}$$

#### 4.2.1. Topological Analysis

Figure 6a,b show the 3D phase space of the $ROS_{EUR}$ and $ROS_{EUR\_MOD}$ maps, iterated using fixed-point architecture. There, we used Δt=0.001. Figure 6a shows the phase space of $ROS_{EUR\_MOD}$ map using S(41,38). It can be seen that the phase space does not present significant changes compared to that of the attractor iterated with floating-point arithmetic (Figure 3b). Therefore, the outputs of the proposed PRNG are the $k_{LSB}$ least significant bits of each state variable $u_{t}$ (see Equation (Equation 17)). This is a commonly used procedure in PRNGs [10,47].
(17)$$u_{t}=b_{wl-1}b_{wl-2}\cdots \underset{PRNG\;(K_{LSB}\;bits)}{\underset{︸}{b_{K_{LSB}-1}b_{K_{LSB}-2}\cdots b_{0}}}$$

It is desired to find the minimum *wl* that keeps oscillating the attractor in a non-periodic way, and the largest $k_{LSB}$ that produces the output sequences to pass the Marsaglia and NIST tests. In Figure 6b, it can be seen how the obtained sequence does not present any structure and all the space is equally filled.

#### 4.2.2. Statistical Analysis

Returning to Figure 5 where the causal entropy × complexity plane was shown, the red stars correspond to the $ROS_{EUR\_MOD}$ map using a signed fixed-point architecture with 41 bits, of which 38 are used to represent the fractional part (S(41,38)). It can be seen that the utilization of fixed-point arithmetic does not influence the statistical properties of the proposed system. Finally, when the most significant bits are discarded (pink point), it can be seen that the sequences reach the ideal point in terms of randomness, $H_{BP}=1$ and $C_{BP}=0$. Table 1 shows the results obtained when applying the Marsaglia battery of statistical tests to the original system ($ROS_{EUR}$ map) and the modified one ($ROS_{EUR\_MOD}$ map). It can be seen that the $ROS_{EUR}$ map needs more bits to pass the tests. There, it is demonstrated that the proposed method keeps the system oscillating and enables it to discard fewer bits of each output. This is due to the fact that the time correlations and internal structures are destroyed. To confirm our proposal’s usefulness, we keep $k_{LSB}$ bits of the output of both the original and the modified maps, iterated using S(*wl*,*fl*) arithmetic, and test them using Diehard tests.

Table 1 shows that the sequences generated by $ROS_{EUR\_MOD}$ pass Marsaglia tests using fewer word bits (*wl* = 41) and allows to keep more bits per iteration for the PRNG (higher $k_{LSB}$) than $ROS_{EUR}$ map (*wl* = 56). Table 2 shows the results of testing the proposed PRNG via the NIST SP 800-22 test suite. In agreement with the values used in the literature [13,47,48,49], 1000 sequences of length $10^{6}$ bits each have been tested. For a significance level of 0.01 (α = 0.01) and 1000 samples, the minimum pass rate for each statistical test is approximately 980, with the exception of the random excursion (variant) test where it is approximately 597 for a sample size of 611 binary sequences. The proposed PRNG passes all these tests.

### 4.3. Hardware Implementation

Here we show that the proposed modification is extremely simple to implement and assures the required randomness. Figure 7 shows a schematic of a hardware implementation of $ROS_{EUR}$ and $ROS_{EUR\_MOD}$ maps. In the latter map, the parameters used were $p_{1}=p_{2}=p_{3}=0.5$.

Figure 7a shows a schematic of the recursive function for *x* of a general map obtained by the Euler method applied to a continuous-time chaotic system (recursive functions for *y* and *z* are analog, and for simplicity are not shown). There, the $f_{x}$ block receives the three state variables at time *t* and calculates the next output at time t+1. In the case of the Rössler system studied here, this term is $f_{x}=-y_{t}-z_{t}$. Its output is multiplied by Δt and added to $x_{t}$ in order to generate $x_{t+\Delta t}$. This value is then latched by a register at each clock cycle. Figure 7b shows the proposed modified circuit. It can be seen that it consists of just one extra term. This term makes the product of one state variable (in this case $z_{t}$) by 0.5. However, for its implementation, no multiplier is required since this term is a right-shifted version of the state variable. Then the least significant bit of the integer part of $x_{t}$ is fed back to select if the new term is added or subtracted. The whole term can be implemented by a positive or negative right-shifted version of $z_{t}$, as it can be seen in the light blue square of Figure 7b. The figure shows that it requires very few resources.

A comparison of the implementation results of the proposed PRNG with three other PRNGs can be seen in Table 3. $ROS_{EUR}$ refers to a PRNG based on the $ROS_{EUR}$ map, which basically consists of implementing the map and then applying the discarding method. The map requires a word length of 56 bits to enable the extraction of 38 bits (of each variable) on each iteration. This word length is the minimum number of bits for $ROS_{EUR}$ to pass the Marsaglia test (see Table 1). We can see that this PRNG requires more resources and that the maximum frequency and throughput are lower than that of the proposed PRNG, which is due to the need to use a larger word size to ensure the randomness of the output. In the third column, the resources used by an implementation of the well-known PRNG Mersenne Twister (MT19937) implemented in an FPGA are shown [50]. We can see that the resource requirement is slightly higher than that of the proposed PRNG. The maximum operating frequency is lower and the achieved throughput is lower as well. The last column corresponds to a continuos-time, chaotic-based PRNG that employs a linear feedback shift register to obtain the transition between Lorenz-like and Chen-like behaviors. Then, authors keep the eight least significant bits and xor them (the discarding method) [13]. Although this generator does not comply with being generic (it only works for the Lü-like chaotic system because it is capable of exhibiting both Lorenz-like and Chen-like chaotic system behaviors for different parameter values), we include it in order to compare its performance.

## 5. Conclusions

In this paper, we showed that the digitalization method and the time step have a significant influence on digitalized systems’ dynamics and, therefore, on the sequences generated by them. The chaotic behavior and statistical degree of the sequences were analyzed using tools from nonlinear systems analysis and information theory quantifiers. In that way and with the objective of using these systems as PRNGs, we proposed a modification to the map generated by Euler’s method that destroys the time correlations of the output and keeps the chaotic oscillation. We have also analyzed the randomness behavior with the amplitude discretization using different precision and data widths. The $H_{BP}\times C_{BP}$ plane shows that the three methods of digitalization analyzed produce outputs in almost the same area, poor balance of amplitudes, and strong inner correlations. The proposed method produces both floating and fixed-point architectures moving towards the typical area of the chaotic maps.

Our goal was to demonstrate that our proposed modification to the digitalized Euler system generates the most random output, which is located in the optimum point of $H_{BP}\times C_{BP}$ plane (uncorrelated noise). The proposed modification achieves the lower value of wl and higher value of $k_{LSB}$ when passing the Marsaglia Diehard and NIST tests. Our method is general and can be applied to any continuous-time chaotic system.

Regarding portability and reproducibility, which are important PRNG properties, the proposed schematic defines the architecture and precision of the variables and the internal calculations since it is a hardware implementation. Then, identical results will be obtained in any programmable device, and the repeatability of the results will be ensured.

Further study on the basin of attraction of the digitized system would be required to define the set of available seeds for the PRNG.

Finally, we compared the resources needed to implement our and other existing methods to obtain PRNGs. We showed that the proposed PRNG is superior in terms of resources, maximum frequency of operation, and throughput.

## Figures and Tables

**Figure 1 entropy-23-00671-f001:**
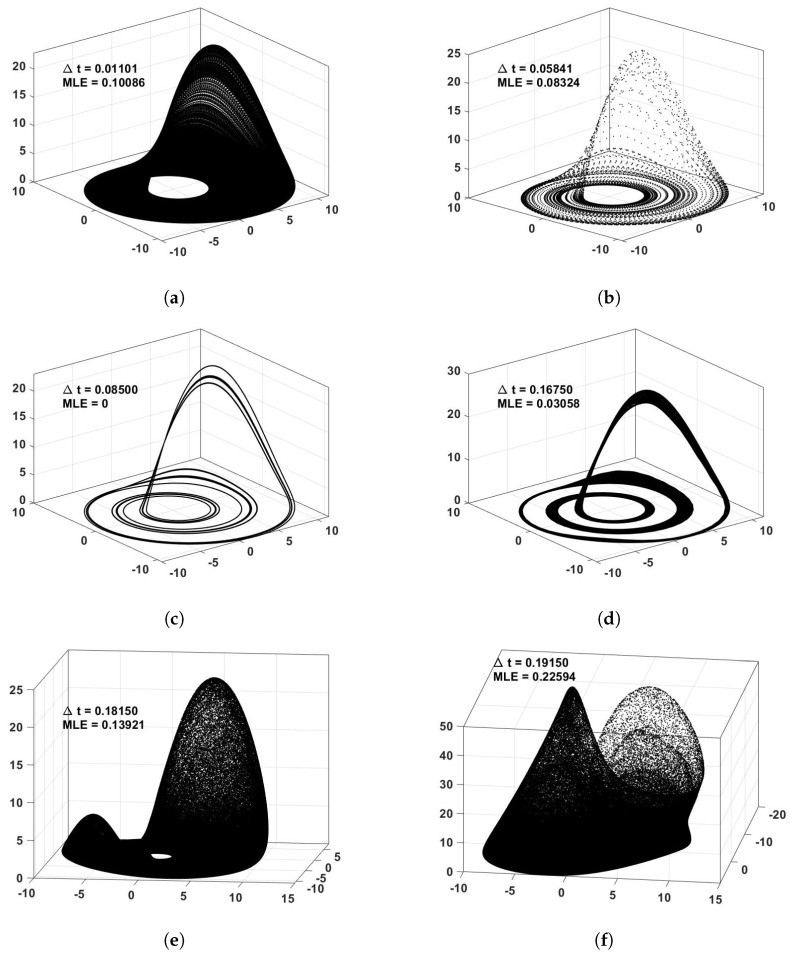
Attractors of the $ROS_{HUN}$ map for different values of Δt and their MLE value (**a**–**f**).

**Figure 2 entropy-23-00671-f002:**
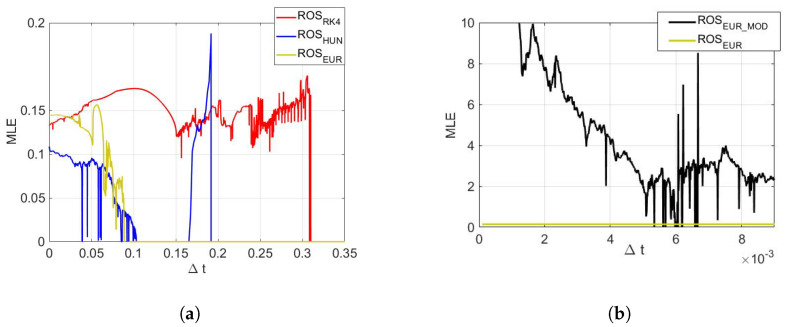
MLE with Δt as a parameter for the Rössler system using different time-discretization algorithms and the proposed method, with floating-point arithmetic. (**a**) MLE for $ROS_{RK4}$, $ROS_{HUN}$, and $ROS_{EUR}$ maps. (**b**) MLE for $ROS_{EUR}$ and $ROS_{EUR\_MOD}$ maps.

**Figure 3 entropy-23-00671-f003:**
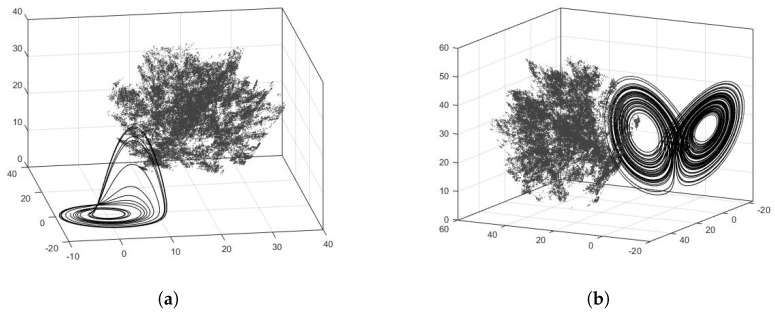
Phase spaces of two well-known continuous-time chaotic systems digitalized by the Euler method and their modified versions. In black are the classical systems, and in gray are the modified versions. It can be seen that, by using the proposed method, the original attractors have been broken and the spaces are more uniformly filled. (**a**) Attractors of $ROS_{EUR}$ and $ROS_{EUR\_MOD}$ maps. (**b**) Attractors of $LOR_{EUR}$ and $LOR_{EUR\_MOD}$ maps.

**Figure 4 entropy-23-00671-f004:**
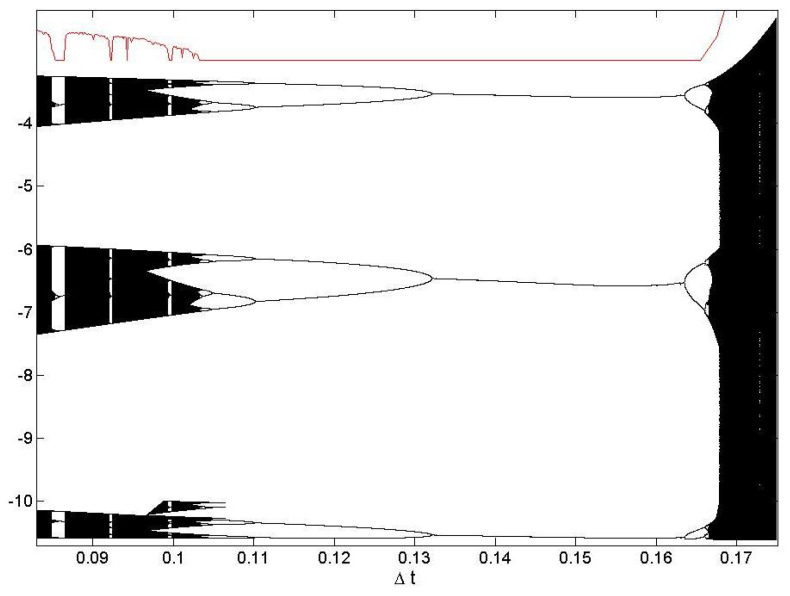
The $ROS_{HUN}$ bifurcation diagram with Δt as a parameter, for the $z_{t}$ variable with the plane $x_{t}=0$ and the MLE superimposed (red line). Note that the y-axis values correspond to the plane, not the MLE.

**Figure 5 entropy-23-00671-f005:**
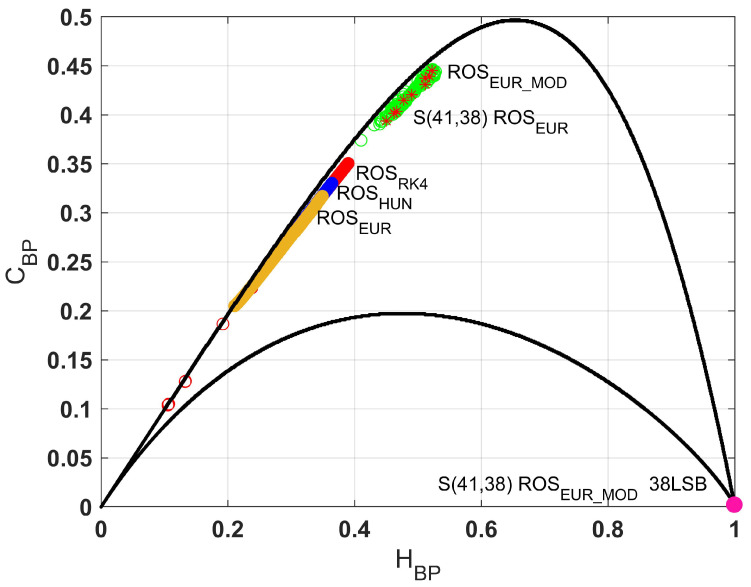
Causal entropy × complexity plane, Rössler system using the three methods of time-discretization for 0.0001≤Δt to the higher Δt that could be reached before the maps diverge. Red points are the $ROS_{RK4}$ map, blue points are the $ROS_{HUE}$ map, and yellow points are the $ROS_{EU}$ map. The $ROS_{EUR}$ and $ROS_{EUR\_MOD}$ maps using S(41,38) are the green points and black stars, respectively. Finally, our proposed PRNG ($ROS_{EUR\_MOD}$ map S(41,38) considering the 38 least significant bits) are the pink points, and these are the best sequences in terms of randomness (closest to the ideal point $H_{BP}=1$, $C_{BP}=0$).

**Figure 6 entropy-23-00671-f006:**
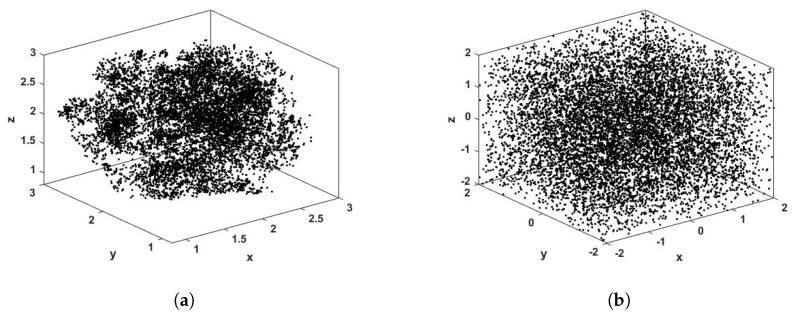
Phase space of the $ROS_{EUR\_MOD}$ system iterated with S(41,38) arithmetic. (**a**) Considering all *wl* bits. (**b**) Considering the $k_{LSB}=38$ bits.

**Figure 7 entropy-23-00671-f007:**
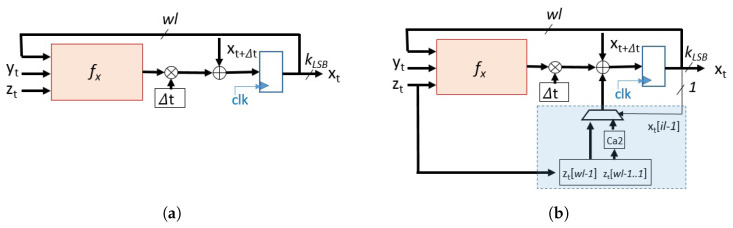
Block diagram of hardware implementation of the maps, in this case, $f_{x}=-y_{t}-z_{t}$. (**a**) $ROS_{EUR}$ map. (**b**) $ROS_{EUR\_MOD}$ map.

**Table 1 entropy-23-00671-t001:** Results from the Marsaglia Diehard test for $ROS_{EUR}$ and $ROS_{EUR\_MOD}$ maps for different precision using S(*wl*,*fl*) architecture and considering the 38 least significant bits ($k_{LSB}=38$) and Δt = 0.001.

*wl*	*fl*	${ROS}_{EUR}$	${ROS}_{EUR\_MOD}$
40	36	fail	fail
40	38	fail	fail
41	38	fail	success
42	38	fail	success
50	45	fail	success
51	45	fail	success
52	45	fail	success
53	45	fail	success
54	45	fail	success
55	50	fail	success
56	50	success	success

**Table 2 entropy-23-00671-t002:** Results from the SP 800-22 test for the $ROS_{EUR\_MOD}$ map using S(*41*,*38*) architecture and dismissing the 3 most significant bits ($k_{LSB}=38$).

Statistical Test	*p*_Value	Proportion	Result
Frequency	0.060875	980/1000	success
BlockFrequency	0.000163	984/1000	success
CumulativeSums	0.008753	981/1000	success
Runs	0.002993	987/1000	success
LongestRun	0.141256	988/1000	success
Rank	0.961869	986/1000	success
FFT	0.424453	990/1000	success
NonOverlappingTemplate	0.697257	989/1000	success
OverlappingTemplate	0.319084	984/1000	success
Universal	0.116065	990/1000	success
ApproximateEntropy	0.894918	991/1000	success
RandomExcursions	0.330947	603/611	success
RandomExcursionsVariant	0.401777	599/611	success
Serial	0.205531	986/1000	success
LinearComplexity	0.971006	988/1000	success

**Table 3 entropy-23-00671-t003:** Summary of resources for $ROS_{EUR}$ and $ROS_{EUR\_MOD}$ maps using S(41,38) arithmetic.

Resources	${\mathrm{ROS}}_{\mathrm{EUR}\_\mathrm{MOD}}$	${\mathrm{ROS}}_{\mathrm{EUR}}$	MT19937 [50]	Chaotic-Based [13]
Platform	Xilinx Zynq-7000	Xilinx Zynq-7000	Xilinx XCV2000E	Altera EP3C16F484C6
LUT	508	604	539	1826
FF	123	200	660	1826
DSP	20	34	0	0
16-Kbit BRAM	0	0	2	0
$f_{max}\left[\mathrm{MHz}\right]$	50	40	24.234	30.98
Throughput [bits/sec]	$5.7\times 10^{9}$	$4.5\times 10^{9}$	$24.16\times 10^{6}$	$247\times 10^{6}$

## Data Availability

The datasets generated during and/or analysed during the current study are available from the corresponding author on reasonable request.

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
