# Peer review of "From Continuous-Time Chaotic Systems to Pseudo Random Number Generators: Analysis and Generalized Methodology"

_entropy, 2021, doi:10.3390/e23060671_

Round 1

Reviewer 1 Report

This is a very interesting work. I have marked minor details to be fixed in order to improve the clarity.

I have two requests to enhance the work:
1) PRNGs are required to be portable (they produce the same results in different machines) and reproducible (they produce the same sequence given the same initial configuration). Please add these properties to your discussion.

2) It would be very interesting if the authors could compare the running time for producing sequences of their proposal concerning, say, the Mersenne-Twister PRNG, or any other good-quality PRNG running in C, C++, Java or Fortran.

Reviewer 2 Report

In this paper, the authors propose a new discretization method and explore the time-digitalized chaotic system. The chaotic maps generated by discretization methods are analyzed using tools from nonlinear systems analysis and information theory quantifiers.

Comments:

  1. The authors are encouraged to briefly review some discrete systems in Introduction, for examples, 10.1109/TIE.2020.3022539, 10.1109/TII.2020.2992438, and 10.3390/e22101119. Besides, comparisons between continuous systems and discrete systems are also suggested to add in the revised paper.
  2. For a continuous system, the simulation results obtained by the ode algorithm can keep consistent with the results obtained by the hardware experiment of the equivalent circuit. Can the proposed method also meet this?
  3. Figs 1, 3 and 4 lack coordinate axis notation. Moreover, for ROSEUR_MOD map, the results presented are deficient in Section 4.1.
  4. The comparison results given in Table 1 are insufficient, and what are the results obtained by using the other two discretization methods in Section 2?
  5. The authors are suggested to add the hardware experimental results.

Reviewer 3 Report

While I am not an expert in Chaotic systems, I believe the authors have a strong understanding of the field and the results of their work are very convincing.

Being a PRNG, it is a bit unclear to me what the "seed" for the proposed PRNG is, how many bits is it stored on, etc. Perhaps the authors could specify this more clearly.

I have done extensive testing of RNGs, so I can say that the statistical testing is sound; in Table 2, though, the p-values are listed for each test of the NIST STS battery; the authors must specify which test was that? (uniformity of p-values ?)

As a final suggestion, perhaps a comparative table, showing how the proposed PRNG compares to other similar approaches, if there are any (and I understand from the introduction that there are) should be added.

Regarding the English, minor corrections are needed; ex: "There, it was used ..." --> "There, we used...", etc.

Reviewer 4 Report

I have several methodological concerns regarding this study as outlined in my comments below.

  1. Discuss the limitations of related works and suggest how your approach overcomes these limitations. What is the knowledge gap being filled by this article?
  2. What is the difference with Random Number Generator Based on Fractional Order Chaotic System
  3. The NIST tests are not done correctly. You should test more sequences and account for the maximum number of binary sequences that are expected to be rejected at the chosen significance level. The procedure is described in “Randomness Testing of the Advanced Encryption Standard Candidate Algorithms”, see p. 4 and Table 3. For example, if the sample consists of 128 sequences, the rejection rate should not exceed 4 sequences.
  4. The computational complexity of the proposed method should be evaluated.
  5. Discuss the limitations of the proposed method.

Round 2

Reviewer 1 Report

The authors successfully addressed all my comments and suggestions. I did not know there was a hardware implementation of the Mersenne-Twister PRNG. I am thankful for letting me know this. Choosing this implementation for comparison was a much better choice than the one I suggested.

Author Response

We appreciate the comments of the reviewer.

Reviewer 2 Report

The paper focuses on developing a methodology to extract PRNGs from continuous-time chaotic systems. Based on proposed Euler modified method, the ROSEUR_MOD map is constructed to obtain the proposed PRNG. In fact, for the discrete map itself, many new and improved maps have been proposed to generate more complex chaotic sequences, which also produce better PRNGs. The authors present a new method, however, other related research in bibliography need to be briefly described and added to the introduction, which makes the introduction more complete.

Author Response

A paragraph of the introduction has been modified to include more references and highlight the advantages and differences with our method. It should be noted that our method is general and can be applied to any  continuous-time chaotic system, the Rossler system is taken only as an example.

" A successful technique in obtaining random outputs from continuous time chaotic systems is the called Discarding method ( [17,18,19,20] called deep-zoom method in the latter). It basically consists of dismissing the most significant bits of each output, it exploits the fact that chaos analytically relies on the infinitesimal depth of precision digits used. However, to maintain chaotic oscillation they are forced to use a high number of bits (even floating point arithmetic), and complex temporal discretization methods. Furthermore, due to the internal and temporal correlations of these systems, at each iteration a low number of bits for the PRNG can be taken (lacks in speed).

Reviewer 3 Report

I am satisfied by the author's response to my observations.

Author Response

(The authors gave the same response as above.)

Reviewer 4 Report

The changes made are suitable and appropriate. The structure of the study has turned into a more appropriate form.

It was evaluated that it might be appropriate to make a comparison for the advantages and disadvantages of proposed methods by reviewing the RNG structure of following study, which was recently published in the Symmetry.

SIEA: Secure Image Encryption Algorithm Based on Chaotic Systems Optimization Algorithms and PUFs. Symmetry 2021, 13, 824. https://doi.org/10.3390/sym13050824

Author Response

In [8] authors propose a hybrid generator architecture of a cryptographic key generator algorithm and its practical application. They argue that chaotic systems (like radioactive decay and noise in electrical circuits) present difficult predictability but do not show good statistical properties. For this reason, the unpredictable requirements of the proposed generator architecture are there met by chaotic systems and PUF structures. In order to improve the statistical properties, the initial conditions and control parameters of chaotic systems were improved with optimization algorithms, while hash functions were applied to PUF outputs. In the work, they mention that it is possible to use different chaotic systems (discrete-time chaotic system, continuous-time chaotic system, hyperchaotic system, and fractional-order chaotic system). However, in the work they just show results for the discrete-time logistic map, they do not specify which architecture has been used to represent the values, neither the number of bits employed. In the Conclusions section, they comment that the issue of the computing realization of chaotic systems is critical, especially considering the problem of digital deterioration and that they have not addressed it in their work. So they have not addressed the main focus of our work. That is the reason we believe that making a comparison with this work in addition to being unfair would not make sense.

However, we have added a comment about this proposed recent published paper  [8] in the Introduction. Note that, “The use of chaotic systems as Pseudo-Random Number Generators has recently grown because of the multiple advantages they present over     stochastic algorithms [4–7]. In [8] the authors propose a five-step encryption algorithm. One of these parts is a chaotic systems module,  where the system chooses between different  number generators. In the conclusions, the authors highlight the importance of its randomness and present  digital degradation as a subject to study.”  

[8] Muhammad, A. U. S., & Özkaynak, F. (2021). SIEA: Secure Image Encryption Algorithm Based on Chaotic Systems Optimization Algorithms and PUFs. Symmetry13(5), 824.